# Interferon Is Superior to Direct Acting Antiviral Therapy in Tertiary Prevention of Early Recurrence of Hepatocellular Carcinoma

**DOI:** 10.3390/cancers12010023

**Published:** 2019-12-19

**Authors:** Wei Teng, Wen-Juei Jeng, Hwai-I Yang, Wei-Ting Chen, Yi-Chung Hsieh, Chien-Hao Huang, Chen-Chun Lin, Chun-Yen Lin, Shi-Ming Lin, I-Shyan Sheen

**Affiliations:** 1Department of Gastroenterology and Hepatology, Chang Gung Memorial Hospital, Linkou Medical Center, Taoyuan City 333, Taiwan; b101090023@gmail.com (W.T.); weiting1972@gmail.com (W.-T.C.); cutebuw@yahoo.com.tw (Y.-C.H.); huangchianhou@gmail.com (C.-H.H.); lincc53@gmail.com (C.-C.L.); lsmpaicyto@gmail.com (S.-M.L.); happy95kevin@gmail.com (I.-S.S.); 2Institute of Clinical Medicine, National Yang-Ming University, Taipei 112, Taiwan; hwaii.yang@gmail.com; 3College of Medicine, Chang Gung University, Taoyuan City 333, Taiwan; 4School of Traditional Chinese Medicine, College of Medicine, Chang Gung University, Taoyuan City 333, Taiwan; 5Genomics Research Center, Academia Sinica, Taipei 115, Taiwan

**Keywords:** antiviral treatment, early recurrence, propensity score matching

## Abstract

The elimination of chronic hepatitis C infection (CHC) by pegylated interferon plus ribavirin (Peg-IFN/RBV) decreases hepatocellular carcinoma (HCC) recurrence rate. However, the tertiary prevention of HCC recurrence by direct acting antiviral agents (DAA) remains controversial. This study aims to compare the tertiary prevention effect between DAA and Peg-IFN/RBV in CHC-HCC patients. Three hundred and one patients who received curative HCC treatment were retrospectively recruited. The recurrence incidence rate (IR) was compared among patients either receiving Peg-IFN/RBV or DAA regimen or untreated by three timeframes (I: from HCC treatment to antiviral therapy; II: during antiviral therapy; III: after antiviral therapy). The prevention effect between Peg-IFN/RBV and DAA were compared in frame II and III after propensity score matching (PSM) with age, tumor staging, HCC treatment modality, and cirrhotic status. Before PSM, the recurrence IRs in three arms were comparable in frame I, while being lower in the Peg-IFN/RBV and DAA arm compared to the untreated arm in frame II. In frame III, the tertiary prevention effect lasted in the Peg-IFN/RBV arm (*p* < 0.001), but diminished in the DAA arm (*p* = 0.135) compared to untreated patients. After PSM, the HCC recurrence IR was higher in the DAA arm than the Peg-IFN/RBV arm in frame II (2724 vs. 666 per 10^4^ person-years, log-rank *p* = 0.042) and III (5259 vs. 3278 per 10^4^ person-years, log-rank *p* = 0.048). Preantiviral ALBI grade therapy is the only predictor for postantiviral therapy HCC recurrence. In conclusion, the tertiary prevention effect of HCC recurrence was not durable in DAA-treated patients, but persisted in Peg-IFN/RBV treatment patients.

## 1. Introduction

Hepatocellular carcinoma (HCC) is the fourth leading cause of cancer related deaths in the world [1]. Chronic viral hepatitis infection (e.g., HBV, HCV) accounts for nearly 80% etiology of HCC [2]. Successful eradication of chronic HCV infection with the pegylated interferon-based regimen (Peg-IFN) has proved to reduce the occurrence (“secondary prevention”) [3] and recurrence of HCC (“tertiary prevention”) [4,5,6]. However, only selected patients are treated by Peg-IFN-based therapy due to its side effect and contraindication in those with decompensated cirrhosis. The sustained virologic response (SVR) rate in cirrhotic patients is also not satisfactory, which can be as low as 38% [7]. Recently, the discovery of direct acting antiviral agent (DAA) regimens overturns the obstacles encountered in Peg-IFN/RBV regimen, leading to nearly 100% SVR rate and applicable regardless of cirrhotic status [8,9]. However, a recent meta-analysis concluded that the HCC occurrence and recurrence rates were comparable between DAA- and IFN-treated patients regardless of the higher SVR rate of the DAA regimen [10]. Moreover, several reports with single-arm DAA-treated HCV-HCC patients declared abruptly increased HCC recurrence rate after antiviral treatment [11,12,13,14], while conflicting results have been reported in other studies comparing to the untreated [15,16,17,18,19,20,21,22,23,24,25] or IFN-treated groups [10,21,26,27,28,29,30]. Concerns have been raised about the benefit and the timing of adjuvant DAA-based therapy. Some of these studies consisted of an IFN-containing DAA regimen [27,28] and did not adjust for baseline characteristics before comparison [26,30]. The definition of follow-up duration varies across these studies [10,21,23,24,25,26,27,28,29,30] and may lead to inconsistent or conflicting comparison results [31]. We therefore conducted this nested case-control study, using propensity score matching (PSM) to adjust for the confounders, and with application of different follow-up time frames to investigate if the tertiary prevention effect is different between the DAA and PegIFN/RBV regimens in curative CHC-HCC patients.

## 2. Results

### 2.1. Characteristics of the Peg-IFN/RBV, DAA, and Untreated Arms before Matching

A total of 301 CHC-HCC patients that had received curative HCC treatment were analyzed. There were 56 patients treated with peginterferon α-2a (Pegasys) and 46 treated with peginterferon α-2b (Peg-intron) in the IFN arm, while most patients (*n* = 32) received a sofosbuvir-based regimen in the DAA arm (Daclatasvir/Asunaprevir, *n* = 23; Ombitasvir/paritaprevir/ritonavir, *n* = 20; Elbasvir/Grazoprevir, *n* = 4), as seen in Figure 1.

At the time of HCC diagnosis, the untreated patients were older (*p* < 0.0001), had more Child–Pugh class B status (*p* = 0.0414), had larger tumor size (*p* < 0.0001), and none received surgical resection (*p* < 0.0001) than those in the Peg-IFN/RBV or DAA groups. Patients receiving the DAA regimen had more advanced fibrotic status, including higher FIB-4 score (*p* < 0.0001) and higher ALBI grade (*p* = 0.0060) than the Peg-IFN/RBV or untreated groups. The median follow-up duration from HCC complete response was the longest in the Peg-IFN/RBV arm (91.3 (3.4–224.8) months) followed by the DAA arm (53.4 (4.9–190.2) months) and the untreated arm (35.9 (3.8–100.8) months) (*p* < 0.0001), as seen in Appendix A. 

At the start of antiviral therapy, the DAA arm was older (*p* < 0.0001), had more advanced fibrotic status (*p* < 0.0001), and more proportion of HCV genotype 1 (*p* = 0.0009) than the Peg-IFN/RBV group. The duration from HCC complete response to the start of antiviral therapy was comparable between two groups (median: Peg-IFN/RBV vs. DAA: 6.1 (0.1–110.6) vs. 8.2 (0.4–141.8) months, *p* = 0.5216). All treated patients were followed long enough for evaluating their SVR status. SVR rate was higher in the DAA group than the PegIFN/RBV group (96.1% vs. 57.7%, *p* < 0.0001). The follow-up duration from the start of antiviral therapy to last follow-up was much longer in the PegIFN/RBV arm than in the DAA arm (median: 72.5 (9.7–193.2) vs. 29.3 (4–53.6) months, *p* < 0.0001) because DAA has been reimbursed by the Taiwan National Health Insurance Administration since December 2016, as seen in Appendix A.

### 2.2. The Timing of Recurrence, Incidence Rate, and Recurrence Patterns between Different Arms before Matching

During a median follow-up of 53.6 months from HCC treatment, 135 (44.9%) patients had HCC recurrence and less than one-third *n* = 39, 28.9%) were found before the start of antiviral therapy. Fourteen patients encountered HCC recurrence during antiviral treatment, while the other 82 patients had HCC recurrence from the end of antiviral therapy (EOT) to two-year follow-up. Counting from the start of antiviral therapy, the disease-free interval (RFI) was much shorter in the DAA arm than that in the Peg-IFN/RBV arm (median: 12.9 (95% CI: 6–19.7) vs. 24.8 (95% CI: 7.8–41.7) months, *p* < 0.001). The one- and two-year HCC recurrence rate was higher in the DAA arm than in the Peg-IFN/RBV arm (48.1%, 58.4% vs. 20.6%, 45.4% respectively, Log-rank test, *p* = 0.002), as seen in Figure 2A. In the Peg-IFN/RBV arm, patients with and without SVR had comparable HCC recurrence rate (51.8% vs. 51.2%, *p* = 0.9560). In the DAA arm, although the HCC recurrence rate seems higher in SVR patients, it did not reach statistical significance because only one in the three non-SVR patients encountered recurrence (60.8% vs. 33.3%, *p* = 0.3414). There was no significant difference in the HCC recurrence between the sofosbuvir-based regimens and the others (67.7% vs. 56.5%, *p* = 0.3227).

During Frame I, the HCC recurrence incidence rate was comparable among the Peg-IFN/RBV, DAA, and untreated arms (Peg-IFN/RBV vs. DAA vs. untreated: 1409.7 vs. 1562.8 vs. 1660.0/10^4^ person-years, overall log-rank *p* = 0.564), as seen in Table 1 and Appendix A. The recurrence patterns in Frame I were predominantly intrahepatic metastasis (71.8%), comparable between the Peg-IFN/RBV and DAA arms (64.7% vs. 77.3%, *p* = 0.100), as seen in Appendix A. Seven patients with viable HCC at the start of antiviral therapy were excluded from entering Frames II and III. During Frame II, the untreated arm had a much higher incidence rate of HCC recurrence than the DAA and Peg-IFN/RBV arms, while the Peg-IFN/RBV arm had the lowest incidence rate of HCC recurrence (Peg-IFN/RBV vs. DAA vs. untreated: 1152.5 vs. 3126.2 vs. 6652.7/10^4^ person-years, overall log-rank *p* < 0.001), as seen in Table 1 and Appendix A. During Frame III, the incidence of HCC recurrence increased in both the Peg-IFN/RBV and DAA arms. However, the Peg-IFN/RBV arm still had the lowest incidence rate comparing to DAA and untreated arms and there was no difference between the DAA and untreated arms (Peg-IFN/RBV vs. DAA vs. untreated: 3851.0 vs. 5602.4 vs. 6734.6/10^4^ person-years, overall log-rank *p* = 0.186), as seen in Table 1 and Appendix A. During Frames II and III, although an increased proportion of local recurrence was observed in the DAA arm compared to that in the DAA arm during Frame I (50% vs. 22.7%), and slightly higher than that in Peg-IFN/RBV arm (50% vs. 34%), no statistical significance was reached (*p* = 0.112). Six patients (12%) in the Peg-IFN/RBV arm had distant HCC metastasis, while none had distant HCC metastasis in the DAA arm (*p* = 0.015). The recurrent tumor number and maximum tumor size were comparable between the Peg-IFN/RBV and DAA arms (*p* = 0.803, *p* = 0.88, respectively), as seen in Appendix A. 

### 2.3. Tumor Recurrence after Propensity Score Matching between DAA and PegIFN/RBV Patients

After propensity score matching of age, fibrosis status, HCC staging, and HCC treatment modality with a 1:1 ratio at the time of antiviral treatment commencement, fifty patients from each of the Peg-IFN/RBV and DAA arms were analyzed. The characteristics were comparable between these two groups, as seen in Table 2. The duration from HCC complete response to antiviral therapy was also comparable between the Peg-IFN/RBV and DAA arms (median: 6.1 vs. 8 months, *p* = 0.2398). From the start of antiviral therapy, the Peg-IFN/RBV arm had longer RFI and lower one- and two-year cumulative HCC recurrence rates than the DAA arm (median RFI: 24.1 (95%CI: 1.7–46.4) vs. 12.1 (95%CI: 1.5–22.8), *p* < 0.001; one- and two-year cumulative HCC recurrence rates: 22%, 48% vs. 48%, 58%, respectively, Log rank test, *p* = 0.043), as seen in Figure 2B. In the DAA arm, there was no significant difference in HCC recurrence between the sofosbuvir-based regimens and the others (65.0% vs. 56.7%, *p* = 0.5562).

The incidence rate of HCC recurrence in Frame I was comparable between Peg-IFN/RBV and DAA arms (995.0 vs. 1017.1/10^4^ person-years, log-rank *p* = 0.853), as seen in Appendix A. However, the incidence rate of HCC recurrence was higher in the DAA arm than that in the Peg-IFN/RBV arm in Frame II (2724.4 vs. 665.8/10^4^ person-years, log-rank *p* = 0.042), as seen in Appendix A, and Frame III (5259.4 vs. 3277.6/10^4^ person-years, log-rank *p* = 0.048), as seen in Appendix A and Table 1. 

The one- and two-year cumulative HCC recurrence rates from commencement of antiviral therapy were comparable between the SVR and non-SVR arms (37%, 49% vs. 30%, 64%, *p* = 0.076). In all SVR patients, those treated with PegIFN/RBV had significantly longer RFI and lower one-year and two-year cumulative HCC recurrence rates than those treated with DAA (median RFI: 22.8 (2.8–176.4) vs. 10.3 (0.8–41.8) months, *p* < 0.001; one- and two-year cumulative HCC recurrence rate: 16%, 32% vs. 48%, 56% respectively, Log rank test, *p* = 0.008), as seen in Appendix A. 

### 2.4. Risk Factors for HCC Recurrence 

Before PSM, the only predictor for postantiviral therapy HCC recurrence in Peg-IFN/RBV patients was recurrence HCC history prior to antiviral therapy (HR: 2.584 (95% CI: 1.274–5.243), *p* = 0.0094). For the 56 SVR patients in the Peg-IFN/RBV arm, no predictors for HCC recurrence were found, and HCC recurrence history prior to antiviral therapy [HR: 4.210 (95% CI: 1.307–13.56), *p* = 0.0167] was the only predictor for HCC recurrence in non-SVR patients in the Peg-IFN/RBV arm. In DAA-treated patients, ALBI grade II/III (vs. I: aHR: 2.374 (95% CI: 1.310–4.301), *p* = 0.0041) is an independent predictor for HCC recurrence. However, the HCV genotype’s impact on HCC recurrence was only shown in the DAA group (genotype 2 vs. 1: aHR: 2.828 (95% CI: 1.352–5.913), *p* = 0.0064), but not in IFN therapy, as seen in Appendix A. 

After PSM, 53 HCC recurrent patients were compared to the other 47 nonrecurrence patients. Higher serum total bilirubin level, lower albumin level, higher AFP level, lower platelet count, higher ALBI grade, higher FIB-4, and shorter time to HCV treatment were noted in the recurrent group, as seen in Appendix A. The one- and two-year cumulative HCC recurrence rates, independent predictors for HCC recurrence, were analyzed by Cox regression, and were higher during Frame I for pre-HCC treatment ALBI grade (II/III vs. I: adjusted HR: 2.02 (1.03–3.96), *p* = 0.04) and HCC treatment modality (other curative treatment vs. surgical resection: aHR: 7.97 (1.04–61.35), *p* = 0.046). During frame II, antiviral therapy with DAA was the only predictor for HCC recurrence (DAA vs. Peg-IFN/RBV: aHR: 3.40 (1.60–19.37), *p* = 0.039). During frame III, pre-HCV ALBI grade at the start of antiviral therapy (II/III vs. I, aHR: 2.22 (1.25–3.93), *p* = 0.006) was the only independent predictor for HCC recurrence, while the duration from HCC treatment to antiviral treatment did not reached statistical significance by multivariate Cox regression analysis (>8 vs. ≤8 months: aHR: 0.65 (0.36–1.16), *p* = 0.144), as seen in Table 3. From the EOT, ALBI I had longer RFI and lower one- and two-year cumulative HCC recurrence rate than ALBI II/III (median RFI: 39.5 vs. 13.7 months; one- and two-year cumulative HCC recurrence rate: 32%, 39% vs. 40%, 70% respectively, Log rank test, *p* = 0.014).

## 3. Discussion

HCC recurrence in chronic hepatitis C patients was highly associated with viremia [4]. Tertiary prevention by IFN-based therapy has been proved to reduce tumor recurrence and prolong survival in postsurgical resection HCC patients [4,32,33]. However, conflicting reports have shown uncertainty of the tertiary prevention effects of DAA treatment in HCC recurrence prevention [11,12,15,27,34,35,36]. These inconsistent results might come from a selection bias, since patients eligible for IFN-based therapy were younger and with better liver reserve, which are protective factors for HCC development, than those treated with the DAA regimen [37,38]. In addition, the time-lag bias, i.e., different time frames applied for recurrence analysis, as seen in Table 3, may lead to discrepant result interpretation. When the follow-up period begins from HCC treatment, it consists of a considerable duration prior to the start of antiviral treatment and counts the period lacking antiviral agents, leading to possible underestimation [23,24,26,28,29,30]. Furthermore, two of these studies [26,30] firstly included untreated patients as a control group and compared it to IFN-based and DAA arms. However, not only was the follow-up duration in the DAA arm too short (1.3 years in Minami et al. [26] and 1.5 years in Petta et al. [30]), but there were also significant baseline characteristic differences, including age and tumor burden, among the three groups that could further weaken the power of the conclusions that were made. Therefore, it is more logical and reasonable to adjust for potential confounders contributing to recurrence and to begin follow-up from the start of anti-HCV treatment when discussing the tertiary prevention effect of HCC recurrence by different antiviral therapies. There was only one study, conducted by Kinoshita et al. [27], using this rationale to compare the tertiary prevention effect between IFN-based and DAA therapies. 

The strength of our study is that the median follow-up duration from the start of antiviral therapy was longer than two years in the DAA arm (median: 29.3 months), reflecting the tertiary prevention effect of early HCC recurrence, defined as recurrence events within two years post-HCC treatment [39] by the DAA regimen. The use of propensity score matching to stratify the differences between the Peg-IFN/RBV and DAA arms could avoid confounding factors potentially contributing to recurrence. Another strength of this study is using the time-varying exposure of different time frames, including from HCC treatment to the start of antiviral treatment (Frame I), the period of antiviral treatment (Frame II), and from EOT to two years post-EOT (Frame III). In Frames II and III, during and after antiviral therapy, the incidence rate of HCC recurrence was significantly lower in patients treated with Peg-IFN/RBV than in the DAA group prior to and after PSM. The tertiary prevention effect of DAA was momentary during antiviral therapy (Frame II) and diminished after the end of treatment, as seen in Appendix A. The HCC recurrence rate in the DAA treated arm was not higher than the untreated arm (DAA vs. untreated: 5602.4 vs. 6734.6/10^4^ person-years, log-rank *p* = 0.135), echoing recent studies’ findings [23,24,25]. In contrast, the tertiary prevention effect of the Peg-IFN/RBV regimen was sustained throughout the antiviral therapy and even after EOT, showing a significantly lower recurrence rate than the untreated and DAA arms, as seen in Table 1, Appendix A. This finding was in conflict with the study by Kinoshita et al. [27], which reported no significant difference of tumor recurrence between IFN and DAA groups after matching (2 years: 70% vs. 76%, *p* = 0.68). The possible reason is that their study recruited a mixed population of patients receiving interferon plus ribavirin +/− DAA, interferon, and pegylated-interferon in the interferon arm, explaining the much lower SVR rate (36.5%) in their study than in ours (57.7%), which only consisted of patients treated with Peg-IFN plus ribavirin. The other possibility may that all their HCC patients were treated with RFA and half of the patients received more than two times the treatment before antiviral therapy, while ours consisted of surgical resection, explaining the much higher two-year recurrence rate in that study than in ours (DAA, IFN: 76%, 70% in Kinoshita et al [27]. vs. 58%, 48% in the current study). This also explains why their one-year HCC recurrence rate in the IFN-containing group was much higher than that in previous studies reported from Japan (one-year: 26%–35% vs. 0%–9.5%) [40,41]. Another important difference between Kinoshita et al.’s study and ours is that their median follow-up duration in the DAA arm was much shorter than that in the current study (1.8 vs. 2.5 years) [27], as seen in Table 4. 

The ALBI score, derived from a regression model of albumin and bilirubin values, has been proposed as an objective grading system to evaluate the functional liver reserve in patients with cirrhosis or HCC. It has been widely validated for prediction of treatment outcome in patients with different HCC staging and treatment modalities [42,43,44,45]. Poor liver function, assessed by the ALBI grade, is associated with higher incidence of tumor recurrence [46,47]. In this study, patients with advanced liver fibrosis had higher risk for postantiviral treatment HCC recurrence. Patients of ALBI grade I had a significantly longer RFS rate and lower cumulative recurrence rate than their counterparts. Thus, patients with advanced fibrosis, even with SVR status, still require continued HCC surveillance.

The possible mechanism for the temporary protective effect by DAA compared to the durable effect of Peg-IFN/RBV therapy may due to the different immunological changes after different antiviral treatments. Recent studies showed the loss of intrahepatic immune activation by IFNα after DAA therapy due to the decreased levels of chemokine C-X-C motif ligand 10 (CXCL10), CXCL11, and a rapid decrease in NK cell activation and a normalization of NK cell cytotoxic effector functions [48], leading to ineffective surveillance of neoplastic clones. Another possible explanation is the epigenetic changes that induce H3K27ac modifications by chronic hepatitis C infection, associated with increased HCC risk, which persists even after HCV cure by DAA therapy [49].

There are several limitations in this study: First, the sample size was reduced after propensity score matching and increased censored cases in the DAA-treated arm two years after the end of HCV treatment were noted. Therefore, this study focused on early recurrence (<2 years), and the impact of antiviral therapy on late recurrence (>2 years) cannot be investigated in current setting; Second, TACE was used in nearly one-quarter of enrolled patients, which remains disputable as a curative treatment modality, although complete response has been documented in certain groups.

In conclusion, both IFN and DAA had different durabilities in their tertiary prevention effect for HCC recurrence in CHC-HCC patients after curative HCC treatment. The Peg-IFN/RBV regimen had a much better and more durable preventive effect than the temporary effect of DAA therapy. Close surveillance for detection of HCC recurrence is mandatory in curative HCC patients choosing DAA as a tertiary prevention regimen.

## 4. Materials and Methods

### 4.1. Patient Recruitment

Chronic hepatitis C patients with curative HCC status achieved by resection, radiofrequency ablation (RFA), transarterial chemoembolization (TACE), and/or proton therapy between January 2001 and December 2017 in Chang Gung Memorial Hospital, Linkou branch treated with antiviral agents were retrospectively recruited. Patients who received antiviral therapy and reached SVR before HCC treatment were excluded. Demographic features, including age, gender, fibrosis status, HCC status, HCV genotype, HCV viral load, and HCC treatment modality, were recorded at of the time of HCC diagnosis, HCV treatment initiation, and end of antiviral treatment. All subjects gave their informed consent to receive HCC treatment, and this retrospective study was approved by the Institutional Review Board committee in Linkou Chang Gung Memorial Hospital (201701340B0C501). 

### 4.2. Diagnosis of Hepatocellular Carcinoma and Follow-Up Protocol

HCC was diagnosed with hyperattenuation in the arterial phase and washout in the late phase [50] by multiphasic, contrast-enhanced imaging (computed tomography (CT)/magnetic resonance imaging (MRI) scans) and/or histology according to European Association for the Study of the Liver/European Organization for Research and Treatment of Cancer (EASL/EORTC) diagnostic guidelines [51]. CT/MRI scans were performed approximately one month after HCC treatment to determine complete tumor response and before the initiation of antiviral therapies to confirm the absence of viable HCC nodules. Complete tumor response was defined as the absence of residual tumor or complete necrosis, and the images obtained before 2010 were reinterpreted by a radiologist according to the modified Response Evaluation Criteria in Solid Tumors (mRECIST) [52]. We monitored HCC recurrence by dynamic CT or MRI every 3–4 months for the first two years, which was then extended to six month intervals thereafter, and measurement of serum alpha-fetoprotein (AFP) levels was incorporated. HCC recurrence was diagnosed using the same criteria as applied to the diagnosis of HCC. 

### 4.3. Antiviral Regimens for HCV Eradication

The PegIFN/RBV regimen composed of peginterferon α-2a (180 mg/week) or peginterferon α-2b (1.5 mg/kg/week) subcutaneously plus weight-based ribavirin (1000 mg/day for weight <75 kg and 1200 mg/day for weight >75 kg). Undetectable serum HCV-RNA at 24 weeks after the cessation of treatment was SVR. Patients received DAA therapy according to their HCV genotype and the severity of liver disease, in accordance with the current guidelines [53]. Undetectable HCV RNA at 12 weeks after the end of HCV treatment (EOT) defined as SVR. Those who failed to achieve SVR were defined as treatment failures (TF).

### 4.4. Laboratory Methods

Hemogram and liver biochemistry tests were performed using automated techniques at the clinical pathology laboratories of the hospital. Commercial kits were used for serum anti-HCV assay (Abbott Laboratories, North Chicago, IL, USA) and AFP level (Abbott Laboratories, lower limit of detection: 2 ng/mL). The HCV-RNA levels in this study were measured using a commercial quantitative polymerase chain reaction (PCR) assay, COBAS TaqMan HCV Test (TaqMan HCV; Roche Molecular Systems Inc., Branchburg, NJ, USA, lower limit of detection: 15 IU/mL). The HCV genotype was determined using a genotype-specific probe-based assay in the 5′ untranslated region (LiPA; Innogenetics, Ghent, Belgium).

### 4.5. Statistical Analysis and Definitions

Descriptive data with normal distribution are reported as mean ± standard deviation (SD) or as percentage otherwise as median (range). The independent Student’s *t*-test and Mann–Whitney U test were used to assess differences between groups in normal distributed and non-normal distributed variables, respectively. Chi-square test was used for categorical variables between the 2 groups. Two-tailed *p* value < 0.05 was considered as statistically significant.

To clarify the impact of antiviral regimen on tertiary prevention of HCC recurrence, HCC recurrence rate was calculated according to different time frames: from the end of hepatoma treatment to initiation of HCV treatment (Frame I), the period of HCV treatment (Frame II), and from end of HCV treatment (EOT) to two years post-EOT (Frame III). The corresponding time frames for untreated patients (Control group) were: (Frame I): from end of hepatoma treatment to six months post hepatoma treatment; (Frame II): 6–12 months after the end of hepatoma treatment and (Frame III): 12–36 months after the end of hepatoma treatment, as seen in Figure 3. Disease-free interval (DFI) was defined as the interval between the start of HCV treatment and the time of HCC recurrence or last follow-up. The Kaplan–Meier method and Log-Rank test were applied to estimate the difference of recurrence rate between the Peg-IFN/RBV and DAA arms. The Cox regression model was used to determine the associations of the predictive factors to clinical outcomes. Propensity score matching with a 1:1 ratio was applied to adjust for the differences between the DAA and Peg-IFN/RBV arms, including HCC staging, HCC treatment modality, age, and fibrosis status at time of antiviral treatment initiation. Statistical analyses were performed using SAS version 9.4 and SPSS software, version 20.0 (SPSS, Inc., Chicago, IL, USA).

## 5. Conclusions

Both IFN and DAA had different durabilities in their tertiary prevention effect for HCC recurrence in postcurative HCC treatment of CHC-HCC patients. The Peg-IFN/RBV regimen has a much better and more durable preventive effect than the temporary effect of DAA therapy. Preantiviral therapy ALBI grade is the only predictor for postantiviral therapy HCC recurrence. Close surveillance for detection of HCC recurrence is mandatory in curative HCC patients choosing DAA as a tertiary prevention regimen.

## Figures and Tables

**Figure 1 cancers-12-00023-f001:**
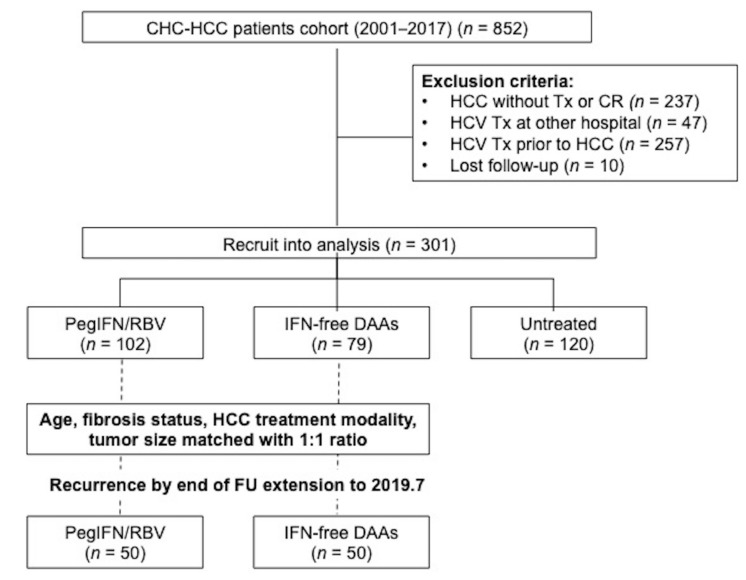
Flowchart of patient recruitment. Abbreviations: CR—complete response; DAA—direct-acting antiviral agents; HCC—hepatocellular carcinoma; HCV—hepatitis C virus infection; PegIFN/RBV—Pegylated interferon plus ribavirin; PSM—propensity scoring matching; Tx—treatment.

**Figure 2 cancers-12-00023-f002:**
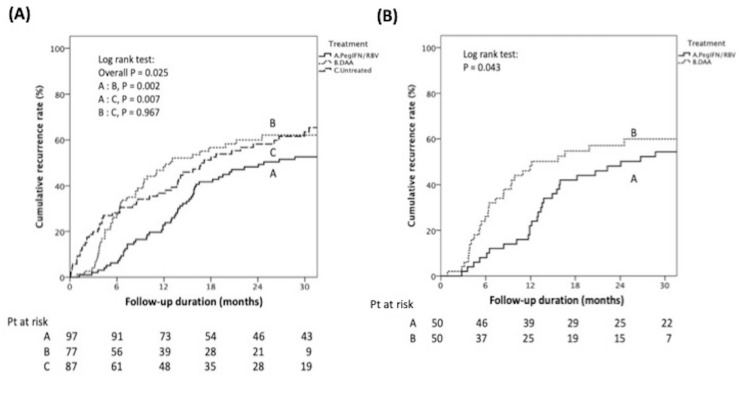
**Kaplan-Meier curve showing cumulative HCC recurrence rate** (**A**) among untreated, Peg-IFN/RBV, and DAA arms before PSM. From the start of HCV treatment, PegIFN/RBV group had the lowest one- and two-year HCC recurrence rate than DAA and untreated group (overall log-rank *p* = 0.025) (**B**) comparison between Peg-IFN/RBV and DAA arms after PSM. From the start of HCV treatment, the PegIFN/RBV group had the lowest one- and two-year HCC recurrence rate compared to the DAA and untreated group (Log-rank *p* = 0.043).

**Figure 3 cancers-12-00023-f003:**
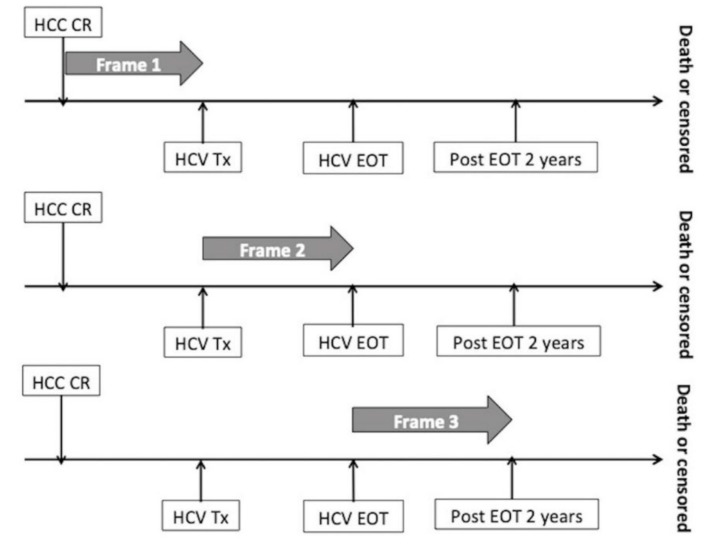
Illustration of different time frames. Abbreviations: CR—complete response; EOT—end of HCV treatment; HCC—hepatocellular carcinoma; HCV—hepatitis C virus infection; Tx—treatment.

**Table 1 cancers-12-00023-t001:** The HCC recurrence rate in different time frames among the untreated, Peg-IFN/RBV, and DAA arms before and after PSM.

Variables	Frame I End of Hepatoma Treatment to Initiation of HCV Treatment	Frame II During HCV Treatment	Frame III EOT to Two Years Post-EOT
Before PSM			
PegIFN/RBV	1409.7/10,000 person-years	1152.5/10,000 person-years	3851.0/10,000 person-years
DAA	1562.8/10,000 person-years	3126.2/10,000 person-years	5602.4/10,000 person-years
	**After the end of hepatoma treatment <6 months**	**After the end of hepatoma treatment 6–12 months**	**After the end of hepatoma treatment 12–36 months**
Untreated	1660.0/10,000 person-years	6652.7/10,000 person-years	6734.6/10,000 person-years
	**Frame I** **End of hepatoma treatment to initiation of HCV treatment**	**Frame II** **During HCV treatment**	**Frame III** **EOT to two years post-EOT**
After PSM			
PegIFN/RBV	995.0/10,000 person-years	665.8/10,000 person-years	3277.6/10,000 person-years
DAA	1017.1/10,000 person-years	2724.4/10,000 person-years	5259.4/10,000 person-years

Abbreviations: CR—complete response; DAA—direct acting antiviral agents; EOT—end of HCV treatment; HCC—hepatocellular carcinoma; HCV—hepatitis C virus; IFN—interferon; PSM—propensity score matching.

**Table 2 cancers-12-00023-t002:** Comparison of patient characteristics between the Peg-IFN/RBV and DAA arms after PSM.

Variables	Overall (*n* = 100)	PegIFN/RBV (*n* = 50)	IFN-Free DAAs (*n* = 50)	*p*-Value
**Time of HCC treatment**				
Age (years) ^†^	63.8 ± 8.0	63.1 ± 7.4	64.5 ± 8.6	0.3847
Gender (male, %)	52 (52.0)	25 (50.0)	27 (54.0)	0.8415
TNM stage I/II/III, *n* (%)	71/26/3 (71.0/26.0/3.0)	37/12/1 (74.0/24.0/2.0)	34/14/2 (68.0/28.0/4.0)	0.7361
HCC treatment, *n* (%) Resection/RFA/Others	24/52/24 (24.0/52.0/24.0)	13/28/9 (26.0/56.0/18.0)	11/24/15 (22.0/48.0/30.0)	0.4337
Total bilirubin (mg/dL)	0.8 (0.3–2.9)	0.8 (0.4–2.8)	0.8 (0.3–2.9)	0.7500
ALT (U/L)	72 (15–372)	73 (15–372)	72 (19–193)	0.8227
Albumin (g/dL) ^†^	3.83 ± 0.49	3.85 ± 0.53	3.80 ± 0.45	0.6095
AFP (ng/mL)	19 (2–6260)	18 (3–4861)	13 (2–6260)	0.3945
Platelet (10^3^/μL)	115 (17–251)	119 (17–251)	103 (33–217)	0.1052
ALBI grade I/II+III, *n* (%)	42/58 (42.0/58.0)	22/28 (44.0/56.0)	20/30 (40.0/60.0)	0.9184
FIB-4	5.17 (1.14–22.3)	5.17 (1.14–19.5)	5.17 (2.16–22.3)	0.2884
APRI	1.97 (0.30–11.7)	1.97 (0.30–11.7)	1.95 (0.52–10.6)	0.6003
Tumor numbers, *n* (%)	1 (1–5)	1 (1–5)	1 (1–5)	0.8131
Target lesion size (cm)	2.0 (0.8–10.0)	2.1 (0.8–10.0)	2.0 (1.0–5.0)	0.2272
**Time of HCV treatment**				
Age (years) ^†^	65.8 ± 8.1	64.9 ± 7.5	66.8 ± 8.6	0.2403
Total bilirubin (mg/dL)	0.8 (0.4–4.1)	0.8 (0.4–2.7)	0.9 (0.4–4.1)	0.2214
ALT (U/L)	86 (16–300)	99 (16–300)	84 (21–198)	0.1329
Albumin (g/dL) ^†^	3.95 ± 0.51	3.91 ± 0.52	4.03 ± 0.49	0.1972
AFP (ng/mL)	14 (2–317)	15 (3–178)	11 (2–317)	0.1242
Platelet (10^3^/μL)	114 (31–251)	114 (57–251)	116 (31–233)	0.5648
ALBI grade I/II+III, n (%)	57/43 (57.0/43.0)	25/25 (50.0/50.0)	32/18 (64.0/36.0)	0.1558
FIB-4	5.63 (0.97–26.1)	6.29 (0.97–18.2)	5.32 (1.39–26.1)	0.8550
APRI	2.44 (0.30–16.0)	2.79 (0.30–16.0)	2.00 (0.45–12.4)	0.2868
Genotype 1/2, *n* (%)	75/25 (75.0/25.0)	31/19 (62.0/38.0)	44/6 (88.0/12.0)	0.0050
SVR, *n* (%)	73 (71.1)	25 (50.0)	48 (96.0)	<0.0001
Time to HCV treatment (months)	7.2 (0.1–141.8)	6.1 (0.1–110.6)	8.0 (0.6–141.8)	0.2398
F/u since HCV treatment (months)	40.1 (11.7–184.1)	74.7 (20.7–184.1)	29.6 (11.7–53.6)	<0.0001

^†^ Demonstrated as mean ± standard deviation. Abbreviations: AFP—alpha-fetoprotein; ALBI—albumin-bilirubin; ALT—alanine aminotransferase; APRI—AST to platelet ratio index; CR—complete response; CTP—Child–Turcotte–Pugh; DAA—direct acting antiviral agents; FIB-4—fibrosis-4; F/u—follow-up; HCC—hepatocellular carcinoma; HCV—hepatitis C virus; PegIFN/RBV—pegylated interferon plus ribavirin; *n*, number; RFA—radiofrequency ablation; SVR—sustained virologic response.

**Table 3 cancers-12-00023-t003:** Independent predictors for HCC recurrence in different time frames after PSM.

Variables		All	Recurrence	Crude HR	95% CI	*p*-Value	Adjusted HR	95% CI	*p*-Value
**Frame 1: End of hepatoma treatment to initiation of HCV treatment**									
HCC Tx modality	Resection	24	1	Referent			Referent		
	Others	76	15	8.510	1.117–64.83	0.039	7.968	1.035–61.35	0.046
TNM stage	I	71	12	Referent					
	II/III	29	4	1.078	0.340–3.422	0.898			
Pre-HCC AFP	≤20	55	8	Referent					
	>20	45	8	1.614	0.600–4.338	0.343			
Pre-HCC ALBI grade	I	42	6	Referent			Referent		
	II/III	58	10	1.755	1.148–4.857	0.041	2.020	1.031–3.956	0.040
HCV treatment	IFN	50	6	Referent					
	DAA	50	10	1.100	0.399–3.033	0.853			
**Frame 2: During HCV treatment**									
HCC Tx modality	Resection	24	2	Referent					
	Others	76	4	0.748	0.136–4.102	0.738			
TNM stage	I	71	4	Referent					
	II/III	29	2	1.470	0.268–8.060	0.657			
Pre-HCV AFP	≤20	61	5	Referent					
	>20	39	1	0.284	0.033–2.437	0.251			
Pre-HCV ALBI grade	I	57	4	Referent					
	II/III	43	2	0.645	0.118–3.522	0.612			
Time to HCV Tx	≤8 months	53	2	Referent					
	>8 months	47	4	2.531	0.463–13.84	0.284			
HCV treatment	IFN	50	2	Referent					
	DAA	50	4	3.397	1.596–19.37	0.039			
**Frame 3: EOT to two years post-EOT**									
HCC Tx modality	Resection	22	8	Referent					
	Others	72	39	1.700	0.826–3.501	0.150			
TNM stage	I	67	31	Referent					
	II/III	27	16	1.571	0.881–2.802	0.126			
Pre-HCV AFP	≦20	56	23	Referent					
	>20	38	24	1.614	0.926–2.814	0.091			
Pre-HCV ALBI grade	I	53	20	Referent			Referent		
	II/III	41	27	2.401	1.366–4.219	0.002	2.217	1.250–3.931	0.006
Time to HCV Tx	>8 months	43	16	Referent			Referent		
	≤8 months	51	31	1.792	1.005–3.195	0.048	1.550	0.861–2.793	0.144
HCV treatment	IFN	48	23	Referent					
	DAA	46	24	1.239	0.707–2.170	0.453			

Abbreviations: ALBI—albumin-bilirubin grade; DAA—direct acting antiviral agents; EOT—end of HCV treatment; HCC—hepatocellular carcinoma; HCV—hepatitis C virus; IFN—interferon; Tx—treatment.

**Table 4 cancers-12-00023-t004:** Summary of currently available studies comparing DAA tertiary prevention effects with untreated or IFN arms.

Source	Year	HCC Patients No.			HCC Tx	HCC Recurrence Rate			F/u Initiation	Median f/u Duration	*p*-Value
		IFN	DAA	Untreated		IFN	DAA	Untreated			
Minami [26]	2016	38	27	861	RFA	1 year: 26.3% 2 years: 52.9%	1 year: 21.1% 2 years: 29.8%	1 year: 30.5% 2 years: 61.0%	HCC Tx	IFN: 3.0 years DAA: 1.3 years Untreated: 3.0 years	0.10
Nagata [28] * (PSM)	2017	22	22	-	Resection RFA	5 years: 54.2%	5 years: 45.1%	-	HCC Tx	IFN: 6.2 years DAA: 2.3 years	0.54
Petta [30]	2017	57	58	328	Resection RFA	5 years: 41.1%	5 years: 39.1%	5 years: 64.5%	HCC Tx	IFN: 2.8 years DAA: 1.5 years Untreated: 1.4 years	0.49
Kinoshita [27] * (PSM)	2018	61	61	-	RFA	1 year: 46% 2 years: 70%	1 year: 51% 2 years: 76%	-	HCV Tx	IFN: 7.2 years DAA: 1.8 years	0.68
Nagaoki [29] (PSM)	2018	32	32	-	Resection RFA SBRT	1 year: 0% 3 years: 34%	1 year: 5% 3 years: 26%	-	HCC Tx	IFN: 5.3 years DAA: 2.8 years	0.36
Huang [23] (IPTW)	2018	-	62	87	Resection RFA SBRT TACE		1 year: 47%	1 year: 49.8%	HCC Tx	DAA: 2.6 years Untreated: 1.8 years	0.93
Singal [24] (PSM)	2019	-	304	489	Resection RFA TACE TARE SBRT	-	aHR: 0.91, 95%CI: 0.69–1.19		HCC Tx	Overall: 0.9 years	>0.05
Cabibbo [25] (PSM/IPTW)	2019	-	102	102	Resection RFA	-	1 year: 15% 2 years: 27%	1 year: 20% 2 years: 40%	HCV Tx	DAA: 1.8 years Untreated: 1.5 years	0.15
Current study (PSM)	2019	50	50	-	Resection RFA TACE Proton	1 year: 22% 2 years: 48%	1 year: 48% 2 years: 58%	- -	HCV Tx	IFN: 6.2 years DAA: 2.5 years	0.04

* IFN-based regimens included PegIFN, RBV, and simeprevir (SMV) or telaprevir (TVR). Abbreviations: DAA—direct acting antiviral agents; EOT—end of treatment; HCC—hepatocellular carcinoma; IFN—interferon; IPTW—inverse probability of treatment weighting; PSM—propensity score matching; RFA—radiofrequency ablation; SBRT—stereotactic body radiation therapy; TACE—transarterial chemoembolization.

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
