# Peer review of "Interferon Is Superior to Direct Acting Antiviral Therapy in Tertiary Prevention of Early Recurrence of Hepatocellular Carcinoma"

_cancers, 2019, doi:10.3390/cancers12010023_

Round 1
Reviewer 1 Report
Overall, this study presented the unsuccessful effects of hepatitis C virus (HCV) DAA drugs on the recurrence of hepatocellular carcinoma (HCC), as compared to that of interferon (IFN) combined with ribavirin (RBV). The authors concluded that HCV DAA treatment could not pronouncedly repress the disease outcome of HCV-related HCC. Before this manuscript is accepted for publication in "Cancer", the data presentation and the significance of this manuscript need to be strengthened by addressing the points as the follows,
The figure resolution of the data presented in this manuscript is not qualified to be published, particularly for the cumulative recurrence rate. The presentation of data in this manuscript must be improved. It is not easy to read this manuscript due to too much number description in the comparison between DAA and IFN/RBV. Several studies have been published for the comparison between the effects of HCV DAA and IFN/RBV on the early recurrence of HCC, such as J. Hepatology 70 (1): 78-86. The conclusion showed that no different rate for HCC recurrence was found between patients treated with HCV DAA and IFN/RBV. Although the authors mentioned the discrepancies between the studies resulted from selection bias, the authors must detailedly compare all the factors that may influence the conclusion between the earlier publication and this manuscript. Is there any difference in the HCC recurrence between different kinds of HCV DAA drugs? The authors must provide current knowledge of HCV DAA used in the clinic and related treatment rationale of these antiviral drugs. The effect of late-stage liver diseases, such as fibrosis and cirrhosis on the recurrence of HCC in patients treated with DAA and IFN/RBV was less discussed in this study.
Author Response
We had modified figure to higher resolution with 2048 pixels width/1536 pixels height and resolution of 600 dpi. The results with simplified data description has been re-written in the manuscript. (line 73-79, page 3; line 134-147, page 4) The different factors that may influence the conclusion between the earlier publication and this manuscript have been re-written. (line 234-273, page 12) There was no significant difference in the HCC recurrence between sofosbuvir-based regimens and others whether before (67.7% vs. 56.5%, P=0.3227) (line 125-126, page 4) or after matching (65.0% vs. 56.7%, P=0.5562) (line 182-183, page 7) The effect of late-stage liver diseases, such as fibrosis and cirrhosis on the recurrence of HCC in patients treated with DAA and IFN/RBV was added in the results and discussion: Results: From the EOT, ALBI I had longer RFI and lower 1- and 2- year cumulative HCC recurrence rate than ALBI II/III (median RFI: 39.5 vs. 13.7 months; 1- and 2- year cumulative HCC recurrence rate: 32%, 39% vs. 40%, 70% respectively, Log rank test, P=0.014).(line 218-220, page 10) Discussion: The ALBI score, derived from a regression model of albumin and bilirubin values, has been proposed to as an objective grading system to evaluate the liver functional reserve in patients with cirrhosis or HCC. It has been widely validated for prediction of treatment outcome in patients with different HCC staging and treatment modalities (reference 40-43). Poor liver function, assessed by the ALBI grade, is associated with higher incidence of tumor recurrence (reference 44-45). In this study, patients with advanced liver fibrosis had higher risk for post anti-viral treatment HCC recurrence. Patients with an ALBI grade I had a significantly longer RFS rate and lower cumulative recurrence rate than their counterparts. Thus, patients with advanced fibrosis even with SVR status still require continued HCC surveillance. (line 367-375, page 14)

Reviewer 2 Report
This manuscript deals with the impact of interferon containing therapy on the recurrence of hepatocellular carcinoma. Therefore, the authors compared patients chronically infected with HCV and curative HCC due to recurrence of HCC dependent on therapy either with or without Peg-IFN alpha. They found that a preventive effect of HCC recurrence is observed in Peg-IFN alpha treated patients but not after DAA therapy. Therefore, the authors conclude that interferon based therapy prevent early recurrence of HCC.
Author Response
Thanks for your precise comments.

Reviewer 3 Report
Overall this is a very important study. The authors compared alpha-interferon to DAA therapy for tertiary prevention of HCC. Pegylated interferon and ribavirin based therapy showed a better preventive effect for the development of HCC. There were some minor issues that should be addressed such as which peg-IFN treatment was used (Peg-Intron vs. Pegasis vs Infergen) as well as which DAA specifically were looked at. Also did the authors evaluate HCC risk in the group of Inferon/ribavirin which did not achieve SVR. The study had a small sample size but this study providers the basis for an expanded study to evaluate HCC development post HCV eradication.
Author Response
There were 56 patients treated with peginterferon α-2a (Pegasys) and 46 treated with peginterferon α-2b (Peg-intron) in IFN arm while most patients (n=32) received sofosbuvir-based regimen in DAA arm (line 64-66, page 3) HCC recurrence history prior to antiviral therapy [HR: 4.210 (95%CI: 1.307-13.56), P=0.0167] is the only predictor for HCC recurrence in non-SVR patients in Peg-IFN/RBV arm (supplementary table 3) (line 201-202, page 10). There was no significant difference in the HCC recurrence between peginterferon α-2a and α-2b whether before (74.1% vs. 69.8%, P=0.6387) or after matching (81.5% vs. 56.5%, P=0.1072).

Round 2
Reviewer 1 Report
All the concerns have been responded in the revised manuscript and rebuttal letter. The reviewer suggests that this manuscript is already prepared to be published.
Author Response
Thanks for your kindly suggestions.
Reviewer 2 Report
The authors made changes according to the mentioned comments unfortunately the response letter misses a point-to-point response which made is difficult to follow the changes.
Page 2, line 48/49: Reference is still missing Question about different genotypes are not answered: The authors mentioned that they found a genotype specific effect on HCC recurrence with non-GT1 being an independent predictor of HCC recurrence. But did the authors could also distinguish between single genotypes (other than GT1 and non-GT1 but GT2, GT3, GT4 and so on). Figures still have low quality/resolution
Author Response
We had added references 8 & 9 in page 2 line 48-49. The majority of the chronic hepatitis C virus infected patients in Taiwan are genotype 1 (around 48%) and genotype 2 (39.5%)[1]. In treated group, our cohort composed of 74.7% genotype 1 and 25.3% genotype 2 CHC patients. However, the HCV genotype’s impact on HCC recurrence only shown in DAA group [HCV genotype 2 vs. 1: aHR: 2.828 (95%CI: 1.352-5.913), P=0.0064] but not in IFN therapy. (Supplementary table 3).(line 161-163, page 10) We had modified figure to higher resolution with 2048 pixels width/1536 pixels height and resolution of 1200 dpi.
References:
Sievert W, Altraif I, Razavi HA, Abdo A, Ahmed EA, Alomair A, et al. A systematic review of hepatitis C virus epidemiology in Asia, Australia and Egypt. Liver international : official journal of the International Association for the Study of the Liver 2011;31 Suppl 2:61-80.
